# Nicotine: From Discovery to Biological Effects

**DOI:** 10.3390/ijms241914570

**Published:** 2023-09-26

**Authors:** Luigi Sansone, Francesca Milani, Riccardo Fabrizi, Manuel Belli, Mario Cristina, Vincenzo Zagà, Antonio de Iure, Luca Cicconi, Stefano Bonassi, Patrizia Russo

**Affiliations:** 1Department of Human Sciences and Quality, Life Promotion San Raffaele University, Via di Val Cannuta 247, 00166 Rome, Italy; luigi.sansone@uniroma5.it (L.S.); manuel.belli@sanraffaele.it (M.B.); mario.cristina@sanraffaele.it (M.C.); antonio.deiure@sanraffaele.it (A.d.I.); stefano.bonassi@sanraffaele.it (S.B.); 2MEBIC Consortium, San Raffaele University, 00166 Rome, Italy; 3Clinical and Molecular Epidemiology, IRCCS San Raffaele Roma, Via di Val Cannuta 247, 00166 Rome, Italy; francesca.milani@studenti.uniroma5.it (F.M.); riccardo.fabrizi@sanraffaele.it (R.F.); luca.cicconi@sanraffaele.it (L.C.); 4Department of Molecular Medicine, University La Sapienza, Viale del Policlinico 155, 00161 Rome, Italy; 5Italian Society of Tabaccology (SITAB), 00136 Bologna, Italy; vincenzo.zaga@icloud.com; 6Experimental Neurophysiology IRCCS San Raffaele Roma, Via di Val Cannuta 247, 00166 Rome, Italy

**Keywords:** nicotine, nicotinic receptor, tobacco, dependence, cell proliferation, neo-angiogenesis, cognition, lung cancer, cardiovascular diseases

## Abstract

Nicotine, the primary psychoactive agent in tobacco leaves, has led to the widespread use of tobacco, with over one billion smokers globally. This article provides a historical overview of tobacco and discusses tobacco dependence, as well as the biological effects induced by nicotine on mammalian cells. Nicotine induces various biological effects, such as neoangiogenesis, cell division, and proliferation, and it affects neural and non-neural cells through specific pathways downstream of nicotinic receptors (nAChRs). Specific effects mediated by α7 nAChRs are highlighted. Nicotine is highly addictive and hazardous. Public health initiatives should prioritize combating smoking and its associated risks. Understanding nicotine’s complex biological effects is essential for comprehensive research and informed health policies. While potential links between nicotine and COVID-19 severity warrant further investigation, smoking remains a significant cause of morbidity and mortality globally. Effective public health strategies are vital to promote healthier lifestyles.

## 1. Introduction

To introduce nicotine and discuss its biological effects, it is necessary to mention tobacco, since nicotine is the psychoactive agent found in tobacco leaves. Currently, there are over one billion smokers worldwide, making tobacco the second most commonly used psychoactive substance [1]. Smokers become addicted to nicotine through the consumption of cigarettes or cigars. This addiction is referred to as tobacco dependence in the International Classification of Diseases, Tenth Revision (ICD-10), or tobacco use disorder in the Diagnostic and Statistical Manual of Mental Disorders, Fifth Edition [2]. Besides nicotine, other components found in cigarettes, such as flavorings and non-nicotine compounds, can influence the addictive potential of tobacco [3].

This review is organized into several sections, beginning with the history of tobacco, followed by a discussion of the characteristics of nicotine, including a botanical overview of the tobacco plant, an examination of nicotine as a secondary metabolite, and concluding with an exploration of the chemical, physical, and biological properties of nicotine. Nicotine is intricate and multifaceted, encompassing historical, pharmacological, biological, and behavioral dimensions. The novelty of this review lies precisely in providing a comprehensive description of nicotine within a single piece of work. The purpose is to evaluate the impact of nicotine as a negative biological agent on human health by examining all its aspects to understand the reasons for its usage and spread. While much is known of the role of nicotine as a psychotropic agent and its impact on neurocircuits [2,4,5,6,7,8,9], its role in human carcinogenesis remains controversial. The availability of data regarding the genotoxic effects of nicotine, including sister chromatid exchange, chromosome aberration, and induction of DNA double-strand breaks in mammalian cells, is not yet sufficient [10,11,12,13]. On the other hand, as outlined in Section 2.4, the role of nicotine in cell proliferation, especially in tumor cells, is well documented, supporting the hypothesis of its role as a promoting agent in the process of human carcinogenesis [14,15,16,17,18,19,20,21,22,23,24,25,26,27,28,29,30,31].

Studies have demonstrated that nicotine alters the expression of microRNAs (miRNAs) in various smoking-related disorders and exerts its effects through miRNA-related pathways [32,33]. In a recent review [32], a comprehensive summary was provided of all the miRNAs influenced by nicotine and the activity of nAChRs. This influence leads to subsequent changes in the expression of target genes. Importantly, alterations in miRNA expression can have both protective effects, such as the activation of anti-inflammatory processes, and detrimental effects, including those associated with conditions like atherosclerosis and Alzheimer’s disease.

Perinatal exposure to tobacco smoke and nicotine has been linked to a multitude of epigenetic changes, such as modified DNA methylation of genes in offspring. These changes are associated with various conditions, including cancer, Alzheimer’s disease, addiction, diabetes, and neural development. This suggests that perinatal nicotine exposure could potentially influence development and increase the risk of disease development through altered DNA methylation patterns. Additionally, prenatal exposure to nicotine has been associated with changes in miRNA signaling linked to inflammatory responses, which are correlated with lower birth weights and disrupted lung development. Both of these factors are connected to developmental exposure to nicotine and tobacco [33].

Developmental exposure to nicotine also leads to changes in histone methylation in the brain and alterations in dendritic complexity, contributing to mental health issues such as depression, addiction, and ADHD. The negative impacts of prenatal nicotine exposure can extend into adulthood, implying that developmental exposure to nicotine can have enduring implications for one’s health.

### 1.1. History of Tobacco

The history of tobacco has its roots in various ancient civilizations. Officially, the recorded history of tobacco is considered to have commenced with the encounter between Christopher Columbus and the indigenous people of the New World in 1492. During their interactions in the Bahamas, the Lucayan, Taíno, and Arawak people presented Columbus and his crew with dried tobacco leaves [34]. However, there are indications suggesting the existence of tobacco even prior to this encounter. In 1976, Michèle (Layer-)Lescot discovered fragments of tobacco leaves in the remains of Ramses II (1279-1213 BCE), the Egyptian pharaoh [35]. Similarly, a German research team reported the identification of psychoactive substances, including nicotine, in Egyptian mummies dating from 1070 BCE to 395 CE [36]. The explanations regarding the presence of tobacco in Egyptian mummies do not account for their post-excavation histories. In fact, the intricate story of the discovery of Ramses II’s mummy involves its movement to various tombs over the course of millennia, introducing the possibility of contamination and intervention.

The consensus among researchers is that tobacco (genus *Nicotiana*) originated in the Andes of South America around 6000 BCE [37]. Cultivated varieties of tobacco, including *Nicotiana rustica* and *Nicotiana tabacum*, characterized by larger leaves and higher nicotine content, spread to regions like Mesoamerica, the Caribbean, and parts of what is now the southeastern and southwestern United States. However, the earliest archaeological evidence comes from a terracotta tobacco pipe discovered in the Banda region of West Africa, dating back to the 19th century BC. This dating is supported by gas chromatography/mass spectrometry (GC–MS) analysis of pipe residues, which identified peaks identical to those found in pure nicotine samples [38,39].

During the first millennium CE, Native Americans began incorporating tobacco into religious ceremonies and for medicinal purposes. The Maya civilization, for instance, utilized tobacco in recreational, ceremonial, and medicinal contexts. They even depicted individuals of high rank smoking cigars, and priests employing tobacco smoke in human sacrifices [40].

The Toltecs, responsible for the establishment of the Aztec empire, adopted the smoking tradition from the Mayans. The Mayans, who inhabited the Mississippi Valley region, introduced the use of tobacco to neighboring tribes, leading to the incorporation of tobacco smoking into their religious rituals. These tribes believed that their deity, Manitou, manifested through the ascending smoke [40].

A Native American myth recounts the tale of a woman dispatched by the Great Spirit to rescue humanity. As she journeyed, wherever her right hand touched the ground, potatoes sprouted, and wherever her left hand touched, corn grew. When she paused to rest, tobacco plants began to flourish, symbolizing the earth’s abundance and fertility [40].

These historical narratives highlight the enduring cultural and ritual importance of tobacco across various civilizations throughout history [41]. In Central America, an intricate system of religious and political practices evolved around tobacco. Over countless years, tobacco has held a revered role within numerous Native American tribes, serving as a conduit for prayer, a symbol of reverence, a source of healing, and a means of protection. The use of tobacco was never meant for misuse and has never been employed for recreational pursuits.

Table 1 presents the history of tobacco.

**Table 1 ijms-24-14570-t001:** History of tobacco.

Years	Note	References
Around 6000 BCE	It is widely acknowledged that tobacco originated and began to grow in the Americas.	[37]
1492	Cristoforo Colombo met Arawaks who offered them gifts, including their much valued dried leaves of tobacco.	[34]
1493	Ramon Pane, a Catholic monk who accompanied Columbus on his second expedition to the West Indies, provided detailed descriptions of the practice of using snuff, a form of smokeless tobacco created by grinding tobacco leaves into a fine powder and typically inhaled through the nose. Pane is commonly acknowledged as the individual who introduced tobacco to Europe during this time.	[41]
1499	Amerigo Vespucci noticed the curious habit of Native Americans chewing green leaves mixed with a white powder.	[42]
1518	Fernando Cortez brought tobacco to Spain, as requested by Ramon Pane.	[4]
1530–1604	The French Ambassador to Portugal, Jean Nicot de Villemain, introduced tobacco to Catherine de Medici, the queen consort and regent of France. Initially, Nicot sent snuff to help treat her son Francis II’s migraine headaches. Later, the queen referred to tobacco as “Herba Regina” or “Herba Medicea”. However, there is some confusion in sources: some assert that it cured Catherine’s own headaches by inducing sneezing.	[4]
1565	Adam Lonicer, Adam Lonitzer or Adamus Lonicerus, in homage to Nicot, named the botanical species *Nicotiana* and its product nicotine.	[4]
1571	Nicolas Monardes, a physician in Seville, wrote a book on the history of medicinal plants of the new world, *De Hierba Panacea*, which stated that tobacco could cure 36 different ills. It became a standard medical textbook across Europe.	[34]
1588	Thomas Harriot, also spelled Harriott, Hariot or Heriot, in Virginia, cited the health benefits of tobacco.	[34]
1600	Tobacco was probably introduced to England by Sir John Hawkins and his crew, although legend reports that it was first used by Sir Francis Drake. There is a legend that Sir Walter Raleigh convinced Queen Elizabeth I to smoke.	[42]
29 October 1618	The tradition that Raleigh smoked a pipe or two on the morning of his execution is well established.	[34]
mid-1660s	When the Great Plague struck London, people believed that smoking tobacco could protect them from infection. (The Great Plague was an outbreak of the bubonic plague that killed more than one-fifth of London’s population.) Eton, a school near the city, even made smoking a requirement in hopes of keeping the plague away.	[42]
1638	Virginia became the primary supplier of tobacco to Western Europe.	[34]
1761	British physician John Hill published *Warnings against Excessive Use of Snuff*, probably the first clinical study of the effects of tobacco. Hill warned snuff users that they were susceptible to nose cancer.	[4]
1776	Tobacco was used by the revolutionaries as collateral for the loans they were getting from France.	[4]
1847	Philip Morris was established in the UK. They were the first to start selling hand-rolled Turkish cigarettes.	[4]
1854–1856	With the return of the veterans from the Crimean War (1854–1856), where soldiers learned to roll tobacco from the Turks, the cigarette spreads throughout Europe.	[4]
1881	Invention of the cigarette-making machine by James Bonsack. He went into business with James ‘Buck’ Duke, and the American Tobacco Company was born.	[4]
1914–1918	The use of cigarettes exploded during the First World War (1914–1918). Everywhere cigarettes were called “soldier’s smoke”.	[4]
1924	Phillip Morris began marketing Marlboro as a women’s cigarette, dubbed “Mild as May!” To combat this publicity, the American Tobacco Company, manufacturer of the Lucky Strike brand, began marketing its cigarettes to women and gained 38% of the market. Lucky Strike urged customers to “Look for a Lucky instead of a sweet”.	[4]
1925–1935	The smoking rate among teenagers triples.	[4]
1940	Americans smoked 2558 cigarettes per person per year, 2.5 times their consumption in 1930; in addition, 7121 cases of lung cancer were reported.	[4]
1941	Ochsner and DeBakey reported a correlation between the increase in tobacco sales and the increasing prevalence of lung cancer, concluding that the latter was mainly due to tobacco.	[4]
1939–1945	During the Second World War (1939–1945), cigarette sales peaked.	[4]
1952–today	Tobacco is classified as a cancer-causing agent due to the presence of various compounds that are considered carcinogenic to humans (at least 70 known to cause cancer, Tobacco-specific nitrosamines (TSNAs), and Polycyclic aromatic hydrocarbons (PAHs))	[4]
2023	Tobacco is responsible for the deaths of up to half of its users. Annually, tobacco claims the lives of over 8 million individuals, which includes 1.3 million non-smokers exposed to second-hand smoke. Approximately 80% of the world’s 1.3 billion tobacco consumers reside in low- and middle-income countries. In the year 2020, the prevalence of tobacco use was 22.3% of the global population, including 36.7% of men and 7.8% of women. To combat the tobacco epidemic, WHO Member States embraced the WHO Framework Convention on Tobacco Control (WHO FCTC) in 2003. Currently, this treaty has garnered the participation of 182 countries. The WHO MPOWER measures are aligned with the WHO FCTC and have demonstrated their effectiveness in saving lives and reducing costs related to averted healthcare expenditures.	[43]

### 1.2. Tobacco Plant

Tobacco is derived from various species of *Nicotiana*, belonging to the botanical family *Solanaceae* (the nightshade family). Among these, *Nicotiana tabacum* stands as the most extensively cultivated species. This plant is identifiable by its short visco-glandular hairs and the release of a yellow secretion containing nicotine [44].

The *Solanaceae* family constitutes a monophyletic group encompassing approximately 99 genera and around 3000 species. This family displays a wide array of diversity in terms of habitats, morphology, and ecology. Although its distribution spans the globe, the highest biodiversity is concentrated in the Americas [45].

The genus *Nicotiana* was named by Linnaeus in tribute to Jean Nicot, a French diplomat who introduced tobacco seeds from Portugal to France in the 16th century [46]. Initially, Linnaeus classified four *Nicotiana* species, all indigenous to the Americas. Later, Lehmann incorporated 21 species originating from Australia. George Don further categorized the family into four sections based on flower shape and color. The taxonomic details of the genus, encompassing the distribution, morphology, and cytology of known species, were meticulously documented in the Goodspeed monograph [47,48]. Goodspeed divided the genus into three subgenera and identified 60 species, including several novel species from Australia, Africa, and South America.

*Nicotiana tabacum* is a perennial or robust annual herbaceous plant that can reach heights of 1–2 m. It features ovate to lanceolate leaves arranged spirally along its stem. *Nicotiana tabacum* is an allotetraploid, likely arising from the hybridization of *Nicotiana sylvestris, Nicotiana tomentosiformis*, and possibly *Nicotiana otophora* [48].

### 1.3. Nicotine as Secondary Metabolite

The general metabolism in an organism includes all metabolic pathways essential for its growth and development. In contrast, specialized metabolites or secondary metabolites (SM) are low-molecular-weight natural products with a narrow taxonomic distribution. They are often synthesized in cells or tissues after active growth has ceased. SMs are typically non-essential for normal growth, development, or reproduction. Their functions, such as pigments and perfumes, include attracting pollinators. SMs encompass a diverse group of natural products synthesized by plants, fungi, bacteria, algae, and animals. They are generally classified into three main groups: terpenes (including plant volatiles, cardiac glycosides, carotenoids, and sterols), phenolic compounds (such as phenolic acids, coumarins, lignans, stilbenes, flavonoids, tannins, and lignins), and nitrogen- or sulfur-containing compounds (such as alkaloids and glucosinolates). SMs play key roles in functions including defense against herbivores and microbial pathogens, UV protection, pollinator attraction, and fertility. They are produced at the highest levels during the transition from active growth to differentiation [49,50].

Nicotine is produced as a defense against predatory insects. Its biosynthesis and aerial accumulation typically increase after herbivore or insect attack, wounding, or jasmonate treatment of the leaf. Experimental evidence supports the hypothesis that tobacco alkaloids, including nicotine, are synthesized in the roots and then transported to the leaves (the site of herbivore or insect attack) through the xylem stream, where they accumulate significantly. In the past, nicotine was used as a pesticide worldwide, including in the United States, until its ban in the mid-1960s [49,50].

During evolution, herbivores and insects develop mechanisms of resistance to nicotine. Notably, the tobacco hornworm (*Manduca sexta*) from the *Sphingidae* family is the only insect that is unaffected by nicotine’s negative effects. Its defense system against nicotine involves carrying an altered amino acid sequence of the receptor, limiting nicotine’s affinity for its receptors, and possessing a functional equivalent of a blood–brain barrier. Astrocytes enveloping neurons express nicotine-binding proteins, acting as scavengers and releasing nicotine into the surrounding hemolymph, protecting the neurons [51]. *Manduca sexta* converts nicotine into metabolites via cytochrome P450 6B46 (CYP6B46), which is known for its unique role in perceiving signaling molecules of plant defense responses [52]. These metabolites are then transported from the gut to the hemolymph, reconverted to nicotine, and released into the air as a deterrent to spiders, termed “toxic halitosis”. However, the braconid wasp *Cotesia congregata* can lay its eggs in the bodies of hornworms, and its larvae feed internally on them, despite *Manduca sexta*’s ability to metabolize nicotine and use it as a defense against predators.

## 2. Nicotine

Nicotine is classified as a tertiary amine consisting of a pyridine and a pyrrolidine ring. It is primarily present in the (S)-nicotine form, and can occur in concentrations as high as 3% in dried leaves of the tobacco plant (*Nicotiana tabacum*). In the lesser-known “Aztec tobacco” (*Nicotiana rustica*), nicotine concentrations can be notably higher, extending to levels as high as 14% [53]. Table 2 provides an overview of the chemical, physical, and toxicological information pertinent to nicotine [54,55,56,57,58].

### 2.1. Nicotinic Acetylcholine Receptors (nAChRs)

Nicotinic acetylcholine receptors (nAChRs) are members of the superfamily of pentameric ligand-gated ion channels, also known as Cys-loop receptors. This name is derived from the presence of conserved residues flanked by linked cysteines at the N-terminal domain of each subunit. These receptors are well conserved from plants to mammals [59,60,61].

Each pentamer of nAChRs consists of an extracellular domain (ECD), a transmembrane domain (TMD) with a central ion channel, and an intracellular domain (ICD). Cys-loop receptors can form both homo- (composed of five identical subunits) and heteropentameric (composed of at least one α and one β subunit) configurations, with the five subunits arranged symmetrically around a central channel axis. Based on their subunit composition and physiological function, nAChRs can be divided into two main classes: muscle type and neuronal type [62].

The International Union of Basic and Clinical Pharmacology Committee on Receptor Nomenclature and Drug Classification (NC-IUPHAR [61]) provides a nomenclature and classification scheme for nAChRs based on the subunit composition of known receptor subtypes [63,64]. Human neuronal nAChRs consist of 11 subunits (eight α subunits: α2–α7, α9–α10; and three β subunits: β2–β4), which generate a limited number of distinct pentameric subtypes. However, the α7 and α9 subunits typically form homopentamers, although α9 may interact with α10 subunits to form heteropentamers (α9–α10). Specifically, in tissues such as the human basal forebrain, α7–β2 heteromers are expressed. The α2–α6 and β2–β4 subunits exclusively form heteromers. All α subunits are involved in forming the ligand-binding site, and at least two α subunits are required for the receptor to be functional [65].

Despite their diversity, all mammalian neuronal nAChR subtypes are permeable to Na^+^, K^+^, and Ca^2+^ ions. nAChR can exist in different conformational states, including closed, open and conducting (activated by ligand binding), and desensitized (closed and unresponsive to ligand binding). The physiological ligand for nAChRs is acetylcholine (ACh). When ACh or nicotine (receptor agonist) binds to the receptor, the ion channel briefly opens, allowing cation flow and altering the membrane potential, typically resulting in depolarization. The channel can then return to its resting state (closed and responsive to activation) or enter a desensitized state, where it is unresponsive to ACh, nicotine, or other agonists [66,67,68].

Although nAChRs are expressed throughout the body, we will focus on their presence in neural and non-neuronal tissues. Neuronal nAChRs are found in nearly every region of the brain, both pre- and post-synaptically, and can be located on axon terminals, axons, dendrites, and somata. On the other hand, non-neuronal nAChRs are expressed on epithelial, endothelial, and immunological cells [69,70,71,72,73].

nAChRs play diverse roles, depending on their tissue location. In neural tissues, they are involved in cognition, addiction, and cell growth. For example, they have been implicated in cognitive processes, addiction-related mechanisms, and cellular growth regulation. In non-neuronal tissues, nAChRs contribute to various functions including inflammation, immunity, and cell growth regulation [69,70,71,72,73].

Furthermore, recent studies have also explored the potential involvement of nAChRs in COVID-19 severity. While the specific mechanisms and implications are still being investigated, there is emerging evidence linking nAChRs to COVID-19 pathophysiology [54,74,75,76,77,78,79].

### 2.2. Nicotine and nAChRs

nAChRs can exist in different conformational states: (i) closed and able to be activated by ligands such as Ach or nicotine; (ii) open and conducting to small cations; and (iii) desensitized, closed, and unresponsive to ligand activation. When ACh or nicotine binds to an nAChR in the open channel state, it rapidly evokes depolarization, allowing cation flux within milliseconds. Subsequently, a gradual decrease in agonist-evoked current indicates channel closure. Prolonged exposure to agonists leads to the desensitization of nAChR, rendering them non-functional [80]. The subunit composition of nAChRs determines the kinetics of these conformational states, the selective cationic permeability of the ion channel pore, and the pharmacological affinities for various agonists. Different nAChR subtypes exhibit distinct functional responses to nicotine. For instance, (α4β2)_2_β2 receptors are considered to be high-affinity receptors, while (α4β2)_2_α4 receptors are classified as low-affinity receptors. Activation of nAChRs can mediate long-term modifications of cellular functions through specific signaling pathways [80,81]. One prominent signaling pathway involving nAChRs, particularly α7 nAChRs, is the generation of complex Ca^2+^-mediated signals. These signals can involve various enzymes and kinases, such as adenylyl cyclase, protein kinase A and/or C, Ca^2+^-calmodulin-dependent kinase, and phosphatidylinositol 3-kinase (PI3K) [82]. In brief, nicotine binding to homomeric (α7 or α9) or heteromeric (α4β2) nAChRs in the concentration range of 10^−8^ to 10^−6^ M leads to the opening of receptor gates, enabling the influx of ions into the cytoplasm. This ion flow induces subsequent membrane depolarization, which then triggers the opening of voltage-gated Ca^2+^ channels. As a result, there is a further elevation in intracellular Ca^2+^ levels. The influx of Ca^2+^ activates downstream signal transduction pathways. In the central nervous system (CNS), both homomeric and heteromeric receptors, when stimulated by nicotine, release DA, contributing to the onset of addiction [80,81,82]. In the case of α7nAChR activation by nicotine, it prompts the release of serotonin, mammalian bombesin, as well as stress neurotransmitters like adrenaline and noradrenaline. However, in non-neuronal cells, these neurotransmitters play a role in fostering the growth of various cancer types. This can occur through direct activation of intracellular signaling pathways (PKC, AKT, ERK) or indirect release of factors that influence proliferation, migration, and angiogenesis (such as epidermal growth factor (EGF) and vascular endothelial growth factor (VEGF)) [80,81,82]. On the other hand, the activation of heteromeric α4β2nAChRs by nicotine in non-neuronal cells prompts the release of the neurotransmitter γ-aminobutyric acid (GABA). Importantly, GABA exhibits a tumor suppressor function for several types of cancer. Interestingly, in neuronal cells, the activation of α4β2nAChRs by nicotine contributes to the development of addiction [8]. This passage highlights the complex and diverse effects of nicotine binding to different nAChRs and the subsequent outcomes on neurotransmitter release, signaling pathways, and cellular responses in both neuronal and non-neuronal cells.

In summary, as reported in the previous paragraphs, AChRs mediate the effects of their physiological agonist, acetylcholine, as well as those of the external agonist nicotine. Dysregulation of AChRs and their downstream signaling pathways can contribute to the development of various diseases.

### 2.3. Nicotine and Biological Effects

Nicotine induces various biological effects, as summarized in Table 3.

The biological effects of nicotine are diverse, and include both negative effects on the cardiovascular system and addiction (now classified as Substance Use Disorders) [2], as well as positive effects such as enhancing cognitive function in individuals with Alzheimer’s disease [86]. A significant portion of the clinical phenotype observed in Alzheimer’s disease (AD) occurs through nAChRs. Degeneration of cholinergic neurons, combined with aberrant nAChR expression and activation partially through amyloid-beta peptide (Aβ)-nAChR leads to the upregulation of pro-inflammatory pathways and subsequently progressive cognitive decline in AD. Interestingly, the cholinergic anti-inflammatory pathway is also mediated through α7-nAChR, in particular. Thus, agonists of these receptors will likely exert pro-cognitive benefits through multiple mechanisms, including stimulating the cholinergic pathway, modulating inflammation, and buffering the effects of amyloid. Despite this promising theoretical use, trials thus far have been complicated by adverse effects or minimal improvement [14,87,88,89].

The most well-known aspect is the involvement of nicotine in addiction phenomena, craving, and reward processes, as discussed in various reviews [4,5,6,7,8,9]. Benowitz, in his seminal review [7] and subsequent works [9,90], provides a comprehensive explanations of these phenomena. In summary, nicotine interacts with nAChR, initiating the release of neurotransmitters—predominantly dopamine (DA), but also norepinephrine, acetylcholine, serotonin, GABA, glutamate, and endorphins—which subsequently induce sensations of pleasure, stimulation, and mood modulation. Activation of these receptors also leads to the establishment of new neural pathways (neural plasticity) and, in conjunction with environmental cues, behavioral conditioning. Following nicotine activation, nAChRs ultimately undergo desensitization, resulting in short-term tolerance to nicotine and diminished satisfaction from smoking. During periods between cigarette consumption or after discontinuing tobacco use, brain nicotine levels decline, causing reductions in DA and other neurotransmitters, accompanied by withdrawal symptoms like cravings. In the absence of nicotine, nAChRs regain their sensitivity to nicotine and are reactivated in response to a new dose.

A new emerging role of nicotine is being observed in relation to human diseases, particularly in the context of COVID-19. Previous studies have demonstrated that nicotine, when present alongside SARS-CoV-2 [54,79], intensifies the cytopathic effects of SARS-CoV-2. This leads to an escalation in the levels of inflammatory cytokines such as TNF-α, IL6, IL8, and IL10, causing significant cellular damage and even cell death. These detrimental outcomes exhibit characteristics akin to pyroptosis and necroptosis. It is worth noting that these severe consequences are notably linked to nicotine’s capacity to activate α7-nicotinic receptors (α7-nAChR), consequently heightening ACE2 activity [78].

Importantly, these effects did not manifest in the presence of an α7-nAChR antagonist (e.g., bungarotoxin) or in cells where α7-nAChR expression was suppressed [54,91]. A systematic review and meta-analysis were conducted to investigate the association between current smoking and the progression of coronavirus disease 2019 (COVID-19). The study analyzed the impact of cigarette smoking on various COVID-19 outcomes, including hospitalization, severity, and mortality. The analysis was conducted up until 23 February 2022. The results of the study suggest that the risk of COVID-19 progressing to more severe conditions and leading to mortality is 30–50% higher for both current and former smokers compared to individuals who have never smoked [77].

In a separate study, Williamson et al. [92] analyzed data from 17,278,392 adults in the UK. They identified a significant risk of death in patients with severe asthma (defined as asthma requiring recent use of oral corticosteroids) and those with respiratory diseases. Patients with chronic obstructive pulmonary disease (COPD) also exhibited worse outcomes upon contracting COVID-19, despite potentially not having a higher risk of contracting the virus initially [92]. Notably, the expression of ACE-2 (the potential receptor for SARS-CoV-2) was notably elevated in COPD patients when compared to control subjects. Additionally, among current smokers, ACE-2 expression was higher than in former and never smokers [78]. These findings strongly indicate that the upregulation of ACE2 resulting from nicotine exposure is contingent upon the activation of α7-nAChR.

Finally, in an Italian population, a borderline significant elevated risk of higher COVID-19 severity was observed among individuals who had ever used e-cigarettes containing nicotine, as compared to those who had never used e-cigarettes (adjusted odds ratio 1.60; 95% confidence interval, 0.96–2.67) [76].

Several natural compounds have been analyzed for their ability to counteract the effects induced by nicotine [93,94,95]. The following compounds appear to be of potential interest:Proanthocyanidins (PCs) and anthocyanins (ACNs) are the predominant flavonoid pigments that are widely distributed in plants, and are known for their therapeutic potential in addressing certain chronic diseases. Treatment with non-toxic concentrations of PCs and ACNs exhibits diverse effects against nicotine-induced non-small-cell lung cancer (NSCLC), encompassing anti-proliferative, anti-migratory, anti-metastatic, anti-invasive, and anti-angiogenic effects, as well as induction of apoptosis and autophagy. The utilization of PC-rich extracts derived from grape seeds and/or Cinnamomi Cortex, in conjunction with radiation or chemotherapy, holds promise for yielding anti-proliferative, anti-inflammatory, and apoptotic benefits against nicotine-induced NSCLC. Moreover, compounds such as delphinidin and cyanidin exhibit the potential to enhance apoptotic and autophagic activity by augmenting the chemosensitivity and/or radiosensitivity of NSCLC cells [93].Quercetin stands as a safe and natural compound with substantial potential to address cigarette-smoking-induced chronic obstructive pulmonary disease (CS-COPD). Quercetin prevents CS-COPD and mitigates airway remodeling through a range of mechanisms, including its antioxidant, anti-inflammatory, and immunomodulatory properties, as well as anti-cellular senescence, modulation of mitochondrial autophagy, and regulation of gut microbiota. Quercetin exhibits potential synergistic effects when combined with beta-agonists and M-receptor antagonists, corticosteroids and roflumilast, antibiotics, and N-acetylcysteine (NAC). This collaboration enhances bronchodilatory, anti-inflammatory, antibacterial, and antiviral effects [94].*Scutellaria baicalensis* and its flavone compounds exhibit therapeutic effects in nicotine-induced NSCLC. These therapeutic effects against NSCLC cells, activated by nicotine via α7nAChR, stem from their capacity to impede proliferation, invasion, migration, metastasis, and angiogenesis. Furthermore, they induce apoptosis, halt cell cycle progression, and trigger autophagy by inhibiting the signaling pathways implicated in NSCLC development. Consequently, targeting α7nAChR and its downstream signaling pathways using flavone compounds holds promise for the development of drugs to counter nicotine-induced NSCLC cells and the treatment of NSCLC in smokers. Combining flavone compounds with chemotherapeutic agents such as cisplatin, which can modulate NSCLC-related signaling pathways, presents a potential strategy for enhancing the anti-NSCLC efficacy of these agents. As such, flavone compounds alone or in synergy with chemotherapeutics could emerge as approved medicinal interventions for NSCLC in smokers [95].

In this review, we will primarily focus on the effects mediated by nicotine binding to α7 nAChR. The α7 nAChR subtype possesses distinctive properties, as outlined in Table 4.

The specific effects induced by nicotine following α7 nAChR activation are diverse when considering human airway epithelial cells, whether tumoral or non-affected. These effects include:Increase in α7 nAChR expression at both mRNA and protein levels [14].Elevation of intracellular calcium ions (Ca^2+^) [14,103].Upregulation of ACE2 expression at both mRNA and protein levels [54,78,91].Activation of signaling cascades such as ERK/MAPK and Phospho-p38 [14,54].Augmentation of proliferation markers like Ki67 and EGFR/EGFR [14,54].Reduction in markers of senescence, such as SA-β-Gal activity and induction of apoptosis markers, including p53/phospho-p53 [54].Induction of EMT (epithelial–mesenchymal transition): decrease in E-Cadherin, increase in Fibronectin (FN), increase in Vimentin [25,54].Increase in markers of neo-angiogenesis, such as VEGF [30,54]. The downstream pathways activated by nicotine, promoting the proliferation, migration, and invasion of airway epithelial cancer cells, as well as of other cancer cell type (i.e., pancreatic [31]). ultimately resulting in a transition toward a more severe neoplastic phenotype.

### 2.4. Ultrastructure of Human Adenocarcinoma Cell Line A549 Treated with Nicotine

This section offers a comprehensive insight into the ultrastructural effects brought about by nicotine, describing the observations made using transmission electron microscopy (TEM) in recent studies. Several studies have explored the ultrastructural alterations induced by nicotine in various types of cell. For instance, in human periodontal ligament stem cells, nicotine was observed to activate α7-nAChR. This activation led to the upregulation of nuclear paraspeckle assembly transcript 1 (*NEAT1*), a nuclear-enriched long non-coding RNA (lncRNA) that plays a critical role as a scaffolding factor for nuclear paraspeckles. Nicotine-induced *NEAT1* upregulation results in the suppression of its functional target gene, *STX17*. This phenomenon contributes to the blockage of autophagy flux and the production of inflammation factors within human periodontal ligament stem cells (PDLSCs) [125]. Given the role played by nicotine in the proliferation of human lung tumor cells, a thorough ultrastructural analysis of the effects caused by nicotine in these cells is important. As a prototype, human adenocarcinoma cells A549 were chosen.

The ultrastructure of untreated human adenocarcinoma cells A549 cells revealed the presence of two distinct cell subpopulations as reported in Sansone et al. [79]. The first subpopulation consisted primarily of well-organized monolayers resembling type 1 pneumocytes. The second subpopulation mainly consisted of cells resembling type 2 pneumocytes, characterized by the presence of basal and apical poles and large osmiophilic lipid granules. These granules contained unsaturated lipid precursors of surfactant and were primarily associated with the Golgi apparatus and endoplasmic reticulum. The mitochondria in the cells appeared well preserved and exhibited a typical orthodox form, with a higher abundance observed in more differentiated cells. The osmiophilic lipid bodies were often closely associated with the apical pole of the plasma membrane, suggesting their involvement in secretion or release into the extracellular space within membrane-bound vesicles [79].

Following nicotine exposure at 0.1 μM for 48 h, A549 cells underwent three main modifications while maintaining their overall preservation and organization [79]. Firstly, the cell area appeared to be substantially increased, with enlarged nuclei and cytoplasm, indicating a hypertrophic variation. Secondly, the cytoplasm contained a higher number of osmiophilic lipid bodies compared to untreated cells. Lastly, the occurrence of cells lacking osmiophilic granules, similar to type 1 pneumocytes, was relatively rare or diminished. Three types of granules were identified: very dense and homogeneous granules, granules with reduced density and homogeneity, and vacuoles containing myelin-like osmiophilic membranes, indicating the final stages of cell differentiation [79]. The majority of cells showed a cytoplasm with a large nucleus. The mitochondria appeared numerous and well preserved. The cytoplasm contained a larger number of osmiophilic lipid bodies. Three types of granules were found: very dense and homogeneous; low density and homogeneous; and vacuoles containing myelin-like osmiophilic membranes [79].

## 3. Discussion

Nicotine is widely acknowledged as the psychoactive substance found in the tobacco plant, which sustains tobacco addiction by binding to nicotinic acetylcholine receptors (nAChRs). This binding process facilitates the release of neurotransmitters such as dopamine, glutamate, and gamma-aminobutyric acid (GABA), thereby mediating the intricate effects of nicotine in individuals who use tobacco [5,6,7,8]. Nicotine has a long history of use, dating back thousands of years before the Common Era (CE). Initially, its use can be traced back to religious rituals and ceremonies. It was often employed in spiritual practices and cultural traditions by indigenous peoples in various parts of the world. Over time, the recreational use of nicotine emerged, as people began to discover its stimulating and mood-altering effects. The consumption of tobacco, which contains nicotine, became popular for its recreational and social aspects (see Table 1). In the years following World War I, a growing body of scientific evidence began to link tobacco use with various health issues, including cancer. As this evidence accumulated, awareness of the harmful effects of tobacco grew, and public health concerns led to increased regulation of tobacco use.

More recently, extensive research has implicated nicotine in a wide range of biological processes, as summarized in Table 4. Figure 1 presents a schematic representation of nicotine’s effects, including its metabolism.

The discovery of nicotine receptors on non-neuronal epithelial cells present in different organs has shed light on the diverse biological effects of nicotine, particularly its role in cell division and proliferation. Moreover, nicotine has been found to stimulate neo-angiogenesis, which refers to the formation of new blood vessels. This process is crucial for tissue growth and repair, but has also been implicated in various pathological conditions, including cancer.

Recent findings suggest a potential link between nicotine and the severity of COVID-19. It has been observed that nicotine, through the activation of τ7-nAChR, can increase the expression of ACE2 (angiotensin-converting enzyme 2). ACE2 is the receptor that the SARS-CoV-2 virus uses to enter human cells. This finding has raised concerns about the potential implications for individuals who smoke tobacco or use nicotine products. If nicotine increases the expression of ACE2, it could theoretically enhance the susceptibility to SARS-CoV-2 infection and potentially worsen the severity of COVID-19 symptoms in smokers [75,76,77,78].

Many smoking individuals suffer from respiratory or cardiovascular diseases. The presence of COPD is common in smokers with frequent exacerbations. These patients, after the acute phase, require rehabilitation programs. These programs should be supported by thorough education about the damage caused by tobacco smoking. Additionally, the effects of nicotine must be well understood in order to achieve an effective rehabilitation process.

Ongoing research is crucial for gaining a comprehensive understanding of the intricate biological effects of nicotine and its potential involvement in various cellular processes and diseases. Further investigation is necessary to fully elucidate the mechanisms and consequences of nicotine’s actions on human health. However, it is important to emphasize that smoking habits are highly dangerous for individuals and public health. With smoking being associated with significant morbidity and mortality worldwide.

Implementing and enforcing effective public health policies is essential for addressing the harmful effects of smoking. This can include the adoption of laws or regulations, the creation of smoking prevention and cessation programs, or the implementation of awareness campaigns.

The enforcement aspect involves the rigorous application of these policies and measures. This means ensuring that smoking-related laws and rules are adhered to and that there are consequences for those who violate them, for example, the enforcement of smoking bans in public places through penalties for anyone who disregards them.

In brief:

Effective public health policies: These are public health policies that are effective at reducing smoking and its related harms. These can include policies that encourage people to quit smoking, reduce access to tobacco products, or promote a healthy lifestyle.

Harmful effects of smoking: The harmful effects of smoking include the negative health consequences associated with smoking, including lung diseases, heart diseases, cancer, and many other health conditions. These harmful effects represent a significant public health threat.

In essence, this study emphasizes that in order to reduce the harmful effects of smoking on people’s health and on society as a whole, it is necessary to have effective public health policies that are rigorously implemented and enforced. These policies can help prevent smoking, promote cessation, and protect people’s health. Furthermore, providing accessible support and resources for individuals looking to quit smoking is crucial for improving public health outcomes.

## 4. Materials and Methods

Major databases, including Medline, Scopus, and Web of Science, were used to discuss:The biosynthesis and accumulation of nicotine in tobacco plants with a focus on its role as a defense against predatory insects.The interaction of nicotine with nicotinic acetylcholine receptors (nAChRs) and its downstream effects.

## 5. Conclusions

Nicotine is a highly addictive and dangerous substance. Governments, healthcare organizations, and society at large should prioritize public health initiatives to combat smoking and reduce the associated risks. By implementing comprehensive strategies, it is possible to protect individuals from the harmful effects of smoking and strive towards a healthier future for everyone.

## Figures and Tables

**Figure 1 ijms-24-14570-f001:**
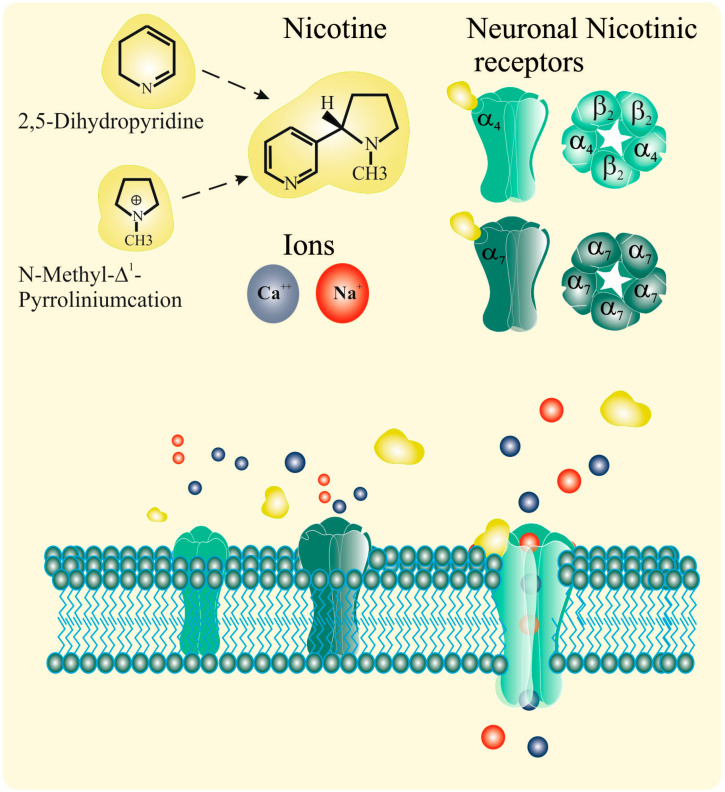
Schematic representation of nicotine’s effects. Nicotine is a tertiary amine consisting of a pyridine and a pyrrolidine ring. (S)-nicotine, which is found in tobacco, selectively binds to nAChRs in a stereoselective manner. The nAChR complex comprises five subunits and is present in both neuronal and non-neuronal cells. There are up to nine α subunits (α2 to α10) and three β subunits (β2 to β4). The most prevalent receptor subtypes in the human brain are α4β2, α3β4, and α7 (homomeric). α4β2* (the asterisk indicates the possible presence of other subunits in the receptor) is the predominant subtype in the human brain, and is believed to be the primary receptor responsible for nicotine dependence. The α3β4 nAChR is thought to mediate the cardiovascular effects of nicotine, while α7 is the most abundant in non-neuronal cells and mediates nicotine’s biological effects, such as cell proliferation, neo-angiogenesis, and resistance to drug-induced apoptosis. Despite their diversity, all mammalian nAChR subtypes are permeable to Na^+^, K^+^, and Ca^2+^ ions. nAChRs can exist in different conformational states, including closed, open, and conducting states (activated by ligand binding such as Ach or nicotine), as well as desensitized states (closed and unresponsive to ligand binding). The physiological ligand for nAChRs is ACh. When ACh or nicotine (a receptor agonist) binds to the receptor, the ion channel briefly opens, allowing the flow of cations and altering the membrane potential, typically resulting in depolarization. The channel can then return to its resting state (closed and responsive to activation) or enter a desensitized state, where it is unresponsive to ACh, nicotine, or other agonists. Nicotine is rapidly and extensively metabolized by the liver, primarily by the liver enzyme CYP2A6 (and to a lesser extent by CYP2B6 and CYP2E1), to form cotinine. Cotinine is subsequently metabolized exclusively or nearly exclusively to trans-3′-hydroxycotinine (3HC) by CYP2A6. The half-life of nicotine averages approximately 2 h, while the half-life of cotinine averages around 16 h. For more comprehensive explanations and references, please refer to the accompanying test.

**Table 2 ijms-24-14570-t002:** Nicotine: chemical, physical, and toxicological data. Adapted from [54].

**Characteristic**	**Description**	**References**
CAS Name	(-)-Nicotine	[54]
CAS Registry Number^®^	54-11-5	[54]
Other Names for This Substance	Pyridine, 3-[(2S)-1-methyl-2-pyrrolidinyl]-. Pyridine, 3-(1-methyl-2-pyrrolidinyl)-, (S)-. 3-[(2S)-1-Methyl-2-pyrrolidinyl]pyridine. (-)-3-(1-Methyl-2-pyrrolidyl)pyridine. Nicotine.	[54]
Chemical Specification and Classes	Bicyclic molecule characterized by a pyridine cycle and a pyrrolidine cycle existing in natures only in the S shape (i.e., levogyre). Biological Agents -> Plant Toxins.	[54]
Physical Description	Nicotine appears as a colorless to light yellow or brown liquid. Combustible. Produces toxic oxides of nitrogen during combustion. Fish-like odor when warm. Chemical formula: C_10_H_14_N_2_, M W: 162.234	[55]
Physical Characteristics	Boiling Point: 247 °C; 125 deg at 18 mm Hg or 476.1 °F at 745 mmHg. Melting Point: −79 °C or 110 °F. Flash Point: 95 °C or 203 °F. Solubility: miscible with water below 60 °C; very sol in alcohol, chloroform, ether, petroleum ether, kerosene, oils. Density: 1.00925 at 20 °C/4 °C or 1.0097 at 68 °F. Vapor Density: 5.61 (Air = 1). Vapor Pressure: 0.038 mm Hg at 25 °C or 1 mmHg at 143.24 °F. Autoignition Temperature: 240 °C or 471 °F.	[55]
SMILES	CN1CCC[C@H]1C1=CC=CN=C1	[55]
IUPAC (InChI)	InChI=1S/C10H14N2/c1-12-7-3-5-10(12)9-4-2-6-11-8-9/h2, 4, 6, 8, 10H, 3, 5, 7H2, 1H3/t10-/m0/s1 check Key: SNICXCGAKADSCV-JTQLQIEISA-N	[55]
Natural Source and Chemical Isolation	Synthesized as secondary metabolite by plants of the family *Solanaceae*, genus *Nicotiana*, species *Nicotiana tabacum*.	[55]
Absorption	Across biological membranes depends on pH. Nicotine is a weak base, with a pKa of 8.0.	[55]
Metabolism	Metabolized in the liver, principally to cotinine, which in turn is metabolized to trans-30 -hydroxycotinine excreted renally. In smokers, more than 90% of the nicotine dose is accounted for by eight metabolites: nicotine N-oxide, nicotine glucuronide, cotinine, cotinine glucuronide, cotinine N-oxide, 3′-hydroxycotinine, 3′-hydroxycotinine glucuronide.	[55]
Human Lethal Dose (LD_100_)	No consensus on the human LD_100_ for adults of 60 mg, resulting in approximately 180 µg L^−1^ plasma concentration.	[56,57]
Nicotine in One Tobacco Cigarette	11.9–14.5 mg of nicotine. On average, a person only absorbs 1–1.5 mg of nicotine from a single stick.	[58]

**Table 3 ijms-24-14570-t003:** Principal biological effects induced by nicotine.

Biological Effects	Comments	References
SUDs: Addictive Properties	Involve the integration of contrasting signals from multiple brain regions that process reward and aversion, attributed to the mesolimbic pathway.	[4,5,6,7,8,9]
Senescence and Atherosclerosis	Nicotine increases MAPK signaling, inflammation, and oxidative stress through NADPH oxidase 1 (Nox1) to induce vascular smooth muscle cell (VSMC) senescence. The accumulation of senescent VSMCs increases the pathogenesis of atherosclerosis by promoting an unstable plaque phenotype.	[83]
Vascular Dysfunction	Nicotine induces vascular remodeling through its effects on proliferation, migration and matrix reduction. Acute effects: myocardial infarction, stroke and sudden cardiac death. Chronic effects: inflammation, thrombogenesis, endothelial dysfunction, hemodynamic stress, arrythmogenesis, insulin resistance and lipid abnormalities.	[84]
COVID-19	Smoking is a potential risk factor for COVID-19 since nicotine upregulates ACE2 expression.	[74,75,76,77,78,79]
Memory and Cognition	Nicotine administration can improve cognitive impairment in Alzheimer’s disease and Parkinson’s disease. Nicotine may also activate thyroid receptor signaling pathways to improve memory impairment caused by hypothyroidism. In healthy individuals, nicotine improves memory impairment caused by sleep deprivation by enhancing the phosphorylation of calmodulin-dependent protein kinase II.	[85,86]
Immune System	Inhibits innate and acquired immunity. Chronic exposure of nicotine plays a critical role in initiating neutrophil recruitment and premetastatic niche formation by skewing neutrophil toward a tumor-supporting phenotype.	[85]
Anti-Inflammatory Function of the Vagus Nerve	Recent studies indicate that the vagus nerve can modulate the immune response and control inflammation through a ‘nicotinic anti-inflammatory pathway’ dependent on the α7 nAChR. Nicotine has been used in clinical trials for the treatment of ulcerative colitis.	[85]
Autoimmune Disease	Multiple sclerosis: nicotine slows down the demyelination process. Rheumatoid arthritis: treatment with nicotine could reduce some of the hematological and biochemical parameters of rats. Sarcoidosis: nicotine increases T.reg levels and decreases TLR2 and TLR9 expression. Inflammatory bowel disease: decreases inflammatory cytokines. Type I diabetes: balance Th1/Th2 ratio. Increases Th1 related cytokines.	[85]
Cancer	Nicotine promotes angiogenesis, proliferation, and epithelial–mesenchymal transition and growth and metastasis of tumors. Different reviews have explored this association in detail.	[14,15,16,17,18,19,20,21,22,23,24,25,26,27,28,29,30,31]

**Table 4 ijms-24-14570-t004:** α7nAChR properties.

α7nAChR	References
*CHRNA7* (Homo sapiens, *NACHRA7*) is located on chromosome 15q12.13	[96]
*CHRNA7* is partially duplicated with *FAM7A* (exons A–E), forming the chimera gene *CHRFAM7A*. Simultaneous transcription of *CHRNA7* and *CHRFAM7A* generates α7 and dupα7 proteins. dupα7 may modulate α7-mediated synaptic transmission or cholinergic anti-inflammatory reaction.	[97,98,99]
*CHRFAM7A* exists in two orientations with respect to *CHRNA7*. Expression of *CHRFAM7A* alone generates protein expression but no functional receptor.	[100]
α7 may be considered a primordial type of receptor because it apparently evolved without additional gene duplications.	[101]
α7 may operate both in ionotropic and metabotropic modes, leading to CICR and G-protein-associated inositol trisphosphate-induced calcium release.	[102]
α7 shows high permeability to Ca^2+^. α7 activates multiple Ca^2+^ amplification pathways. α7 is modulated by extracellular Ca^2+^ concentrations.	[103]
α7 may bind two–five molecules of agonist and modulates cellular functions via phosphorylation and/or via Ca^2+^-dependent serine/threonine kinases.	[103]
α7 is functional without co-assembling with specialized accessory subunits as required by other nAChR subtypes.	[103]
α7 may co-assemble with β2, forming the functional α7β2 receptors expressed in human basal forebrain neurons and cerebral cortical neurons.	[104,105]
Choline is the least potent agonist for α7, with a potency approximately 10-fold lower than Ach. α7 choline-activated current may play an important role in Ca^2+^ homeostasis regulation in α7-expressing cells.	[87,106]
The low probability of α7 being open can be overcome by positive allosteric modulation and serum factors, leading to the generation of excitotoxic currents at physiological temperatures.	[107,108]
The activity of RIC-3 is critical for the folding, maturation and functional expression of nAChR. α7 needs RIC-3 activity for biogenesis and cell-surface expression.	[109,110,111]
α7 requires NACHO, a small multi-pass transmembrane protein enriched in neuronal ER, in combination with RIC-3 for proper assembly.	[112]
Since great amounts of α7 also remain improperly assembled in the presence of RIC-3, it has been suggested that additional chaperone such as cholinergic ligands may promote the α7 assembly.	[113]
α7 is palmitoylated with a stoichiometry of approximately one palmitate/subunit during the assembly in the ER.	[114]
α7 regulates NMDAR, forming a complex α7nAchR/NMDAR via protein–protein interaction.	[115,116]
α7 stimulation is needed for NMDA actions.	[117]
α7 promotes the formation of glutamatergic synapses during development.	[118]
The endogenous “prototoxin” LYNX1, belonging to the Ly6 protein family, binds α7 within the extracellular domain, leaving the classical binding site for agonists and competitive antagonists of α7 nAChR unoccupied.	[119]
The prototoxin SLURP-1 is a positive allosteric modulator of α7.	[120]
α7nAChR is the major nicotinic subtype that is highly expressed in the brain (olfactory bulb, cerebral cortex, hippocampus, hypothalamus and amygdale), as well as in non-neuronal cells (epithelial, immunological, etc.).	[71,121,122,123]
The 3D structure of human α7nAChR is still to be elucidated.	[124]

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
