# Peer review of "Nicotine: From Discovery to Biological Effects"

_ijms, 2023, doi:10.3390/ijms241914570_

Round 1
Reviewer 1 Report
· Numbers are not required in abstract.
· In general, the manuscript is not very well organized, has much less details, and sometimes hard to read. The introduction is very short. The novelty of this review is very week. Why this review is important in light of other reviews? What this review adds? This should be clearly mentioned at the end of introduction. In my view, the review could benefit from expanding the introduction and including extra sections in order convey the key message on the biological significance and biochemical/physiological/pathophysiological effects of nicotine.
· Authors should also consider these questions: Is nicotine a carcinogen or harmful? What are types of nicotine products? What class of drug is nicotine? What carcinogen group is nicotine? How much nicotine is safe? What neurotransmitters other than dopamine are affected by nicotine? How does nicotine affect metabolism? How does nicotine affect epigenetics/DNA methylation and gene expression?
· Several paragraphs in sections 1.1, 1.2, 2.1 and 2.3 are not supported with references.
· Section 2.4: More details on AChRs and its downstream signaling pathways for each disease are required. Tables 3-5 are not very well organized and should include more columns to be clearer.
· Line 136-139: Is there any evidence suggests flavonoids or other phenolic compounds are effective in reducing nicotine-induced diseases? I would suggest autors referring to these articles (ACS Omega. 2021 Oct 5; 6(39): 25361–25371; Int J Mol Sci. 2022 Jul 18;23(14):7905; Ther Adv Respir Dis. 2023 Jan-Dec; 17: 17534666231170800; Int J Environ Res Public Health. 2021 May 14;18(10):5243).
· Section 2.5 is vague. What is the message from this section?
· Materials and methods should be placed as section 2.
· Sections 3 & 5 should be combined into one section entitled “conclusions”.
· Please follow the journal guidelines for referencing.
Moderate editing is required
Author Response
see attachment and the new manuscript

Reviewer 2 Report
The manuscript is a descriptive review on the discovery of Nicotine and its biological effects.
Comments
1. The manuscript is a descriptive review. The Abstract should be written as for a review. The Results do not follow from the methods mentioned. In the current state, it is confusing.
2. The history of nicotine is interesting.
3. Please check all spelling. e.g. COVID-19 not spelled everywhere same.
4. Some sentences are unclear. e.g. in the table of history, Tobacco as a cause of cancer. Here, authors should be more precise. What kind of chemicals in tobacco are carcinogens?
5. Authors should cite some information from WHO.
6. Section 2.5, authors have primarily described one paper. Authors should cover a wider range of papers to describe this issue. Otherwise, change to another title and discuss more widely.
7. Authors may want to include a picture with the chem str of nicotine and how it binds to the receptor, and also metabolism.
8. Tobacco smoke causes genotoxicity. Please mention and cite appropriate papers.
Check spellings in the manuscript.
Round 2
Reviewer 1 Report
Dear Authors,
The paper has significantly improved by these revisions. However, some points remain.
1. Materials and method should be placed as section 2 not section 4.
2. Please avoid numbering order throughout the paper.
3. List of abbreviations must be included at the end after the conclusion section.
4. Tables 2-5 are still not well organized. I would suggest adding additional columns in each table.
5. Line 585: "By implementing comprehensive strategies"- Please expand.
Moderate editing required.
Author Response
see attacment

Reviewer 2 Report
Authors have answered my comments.
Author Response
The reviewer did not request comments. I am attaching the response to reviewer 1.
